# Effect of Lactic Acid Fermentation on Color, Phenolic Compounds and Antioxidant Activity in African Nightshade

**DOI:** 10.3390/microorganisms8091324

**Published:** 2020-08-30

**Authors:** Alexandre Degrain, Vimbainashe Manhivi, Fabienne Remize, Cyrielle Garcia, Dharini Sivakumar

**Affiliations:** 1Phytochemical Food Network Research Group, Department of Crop Sciences, Tshwane University of Technology, Pretoria West 0001, South Africa; alexandre974.degrain@gmail.com (A.D.); vimbainashed@gmail.com (V.M.); SivakumarD@tut.ac.za (D.S.); 2QualiSud, Université de La Réunion, CIRAD, Université Montpellier, Montpellier SupAgro, Université d’Avignon, 97490 Sainte Clotilde, France; cyrielle.garcia@univ-reunion.fr

**Keywords:** traditional leafy vegetables, polyphenols, postharvest processing, FRAP activity, proximate analysis

## Abstract

This study aimed to investigate the influences of fermentation at 37 °C for 3 days by different lactic acid bacterium strains, *Lactobacillus plantarum* (*17a*), *Weissella cibaria* (*21*), *Leuconostoc pseudomesenteroides* (*56*), *W. cibaria* (*64*) or *L. plantarum* (*75*), on color, pH, total soluble solids (TSS), phenolic compounds and antioxidant activity of African nightshade (leaves). Results indicated fermentation with *L. plantarum 75* strain significantly decreased the pH and total soluble solids, and increased the concentration of ascorbic acid after 3 days. *L. plantarum 75* strain limited the color modification in fermented nightshade leaves and increased the total polyphenol content and the antioxidant activity compared to the raw nightshade leaves. Overall, *L. plantarum*
*75* enhanced the functional potential of nightshade leaves and improved the bioavailability of gallic, vanillic acid, coumaric, ferulic ellagic acids, flavonoids (catechin, quercetin and luteolin) and ascorbic acid compared to the other lactic acid bacterium strains. Correlation analysis indicated that vanillic acid and *p*-coumaric acid were responsible for the increased antioxidant activity. Proximate analysis of the fermented nightshade leaves showed reduced carbohydrate content and low calculated energy.

## 1. Introduction

The indigenous leafy vegetable African nightshade (*Solanum retroflexum Dun.*) is a popular food in Venda, Limpopo Province, South Africa. Nightshade leaves are a rich source of the minerals Ca (199 mg/100 g), Mg (92 mg/100 g) and Fe (7.2 mg/100 g), and the raw leaves are rich in rutin [1]. Generally, the consumption of nightshade leaves is in fresh form after cooking, but they have a limited shelf life due to their highly perishable nature. Traditional methods of preserving these indigenous vegetables includes fermentation [2]. Lactic acid bacteria (LAB) are a group of gram-positive bacteria that convert sugars in food into lactic acid. Lactic acid fermentation is one of the oldest forms of food preservation and is affordable at the rural level. Lactic acid fermentation improves the aroma compounds, sensory properties, nutritive compounds (mainly folate, riboflavin, cobalamin, and ascorbic acid), low-calorie polyols (mannitol and sorbitol) and reduces anti-nutritive compounds [3].

Lactic acid fermentation increased the total phenols, flavonoids and antioxidant activity in the indigenous vegetable *Momordica charantia* (bitter melon) [4]. The lactic acid-fermented indigenous fruit camu-camu (*Myrciaria dubia*) and soymilk demonstrated antioxidant and anti-diabetic activities by inhibiting α-amylase and α-glucosidase activity [5]. Fermentation of African nightshade (*S. Scabrum*) using *L. plantarum* BFE 5092 and *Lactobacillus fermentum* BFE 6620 starter strains showed a significant impact on the bacterial composition by reducing the spoilage and pathogenic microorganism populations due to a reduction in pH and production of lactic acid [6].

*L. plantarum* has the “generally regarded as safe” (GRAS) status, warranting the safety of the bacterium for consumption [7]. The LAB strains can have probiotic properties, and the lactic-fermented foods provide beneficial effects on human gastrointestinal health [6].

However, limited information is available on the changes in different phenolic components or antioxidant properties of traditional underutilized vegetables after fermenting with different LAB strains. Furthermore, the use of LAB (e.g., some *L. plantarum* strains) as starter cultures in vegetable fermentation increases the control of fermentation. *Weissella cibaria*, *Weissella confusa* and *Leuconostoc pseudomesenteroides* are used in the food sector because these LAB strains produce exopolysaccharides [7]. *W. cibaria* isolated from fermented vegetables produces bacteriocin and weisseillicin, the natural peptides that can control food-borne pathogens or spoilage microorganisms [8]. Furthermore, the acidification of foods by organic acids produced by LAB stains and bacteriocins during fermentation can extend the shelf life and improve the safety of the product for consumption [9].

In this study, we investigated the influence of five different LAB strains, which had previously been isolated from plant materials and selected for their fermentative and stress resistance characteristics [3], on nightshade leaves during fermentation to select the suitable strains that showed a positive influence on physicochemical properties, phenolic profile and antioxidant property for development of a future functional food. The proximate analysis of the fermented nightshade leaves was included in the study, after selecting the suitable LAB strain.

## 2. Materials and Methods

### 2.1. Chemicals

Acetonitrile, 2,5-dihydroxybenzoic, vanillic, gallic, *p*-coumaric, ferulic, caffeic and ellagic acids, pyrogallol, epicatechin, catechin, quercetin, luteolin (≥95.0–98%), Man Rogosa Sharpe agar and other chemicals were purchased from Merck, Germiston, South Africa.

### 2.2. Plant Material

The irrigation scheme in Thulamela Municipality, Venda, Limpopo Province, harvested the nightshade leaves during the winter season in 2019, as described by Managa et al. (2020) [1]. The leaves were detached from the stem, rinsed in tap water, disinfected (5 min) with NaOCl 5%, rinsed further with distilled water to remove the excess chlorine from the leaves, blanched in a water bath at 95 °C for 5 min and subsequently placed in screw cap bottles (75 g of leaves per bottle) to cool to 30 °C before adding 0.8% NaCl sterile solution.

### 2.3. Starter Cultures and Fermentation of Nightshade Leaves

LAB strains used in the study came from culture collections from University of La Réunion, QualiSud [3]. *L. plantarum 17a* and *75* were isolated from papaya and cabbage respectively. *W. cibaria* 21 and 64 originated from cabbage, as did *L. pseudomesenteroides 56*.

LAB strains were grown on Man Rogosa Sharpe (MRS) agar at 30 °C for 72 h. The strain reactivation involved one or two colonies being suspended in 9 mL of MRS broth, incubated at 30 °C for 48 h and centrifuged at 8000× *g* for 5 min. The cell pellets were washed and rinsed twice with sterile distilled water, and then re-suspended in 20 mL of sterile water to obtain a concentrated cell culture. Thereafter, nightshade leaves, inoculated with LAB (approximately 1 × 10^8^ CFU/mL), were placed at 30 °C, without agitation, for 3 days (duration of fermentation); holding the fermented products at −20 °C for 15 min stopped fermentation. Fermentation was performed in triplicate for each LAB strain.

### 2.4. Physicochemical Properties and Ascorbic Acid Content

All measurements took place on day 3 after fermentation. Total color difference (∆E) was determined using a colorimeter (CR-400 Chroma Meter), according to Managa et al. (2020) [1]. The *L** (lightness), *a** (greenness) *b** (yellowness) color components determined the ∆*E* of each composite sample using the following formula: (1) Fresh leaf samples’ color coordinate values are *L_1_**, *a_1_**, *b_1_** and the samples from fermentation treatments with LAB strains represent *L_2_**, *a_2_**, and *b_2_**
(1)$$\Delta E_{ab}^{*}=\sqrt{{\left(L_{2}^{*}-L_{1}^{*}\right)}^{2}+{\left(a_{2}^{*}-a_{1}^{*}\right)}^{2}+{\left(b_{2}^{*}-b_{1}^{*}\right)}^{2}}$$

The pH of the brine, before and after mixing with leaves (10 mL), was sterilely taken from the bottles, and the pH values recorded using a digital pH meter (Mettler-Toledo Instruments Co., Shanghai, China) at 24 h intervals up to 3 days.

A refractometer measured the total soluble solids (TSS) of the brine solution, after mixing the leaves (Agato pocket PAL-2, Tokyo, Japan) up to day 3, and expressed as %TSS. After testing each sample, the refractometer was calibrated using distilled water.

Ascorbic acid content was determined via titration using 2,6 dichlorophenol indophenol dye, according to AOAC [10].

### 2.5. Quantification of Targeted Phenolic Compounds Using HPLC-DAD and Total Polyphenolic Content

Concentrations of phenolic acids and flavonoids were quantified from snap frozen samples using HPLC analysis, performed using the Shimadzu liquid chromatograph system (Shimadzu Corp, Kyoto, Japan) equipped with a quaternary pump, a vacuum degasser, C18 column and an auto sampler and DAD detector, described by Mpai et al. (2018) [11], without any modifications. The homogenized sample (1 g) was extracted in a solution mixture containing 10 mL of methanol and 1% butylated hydroxytoluene using an ultrasonic bath for 45 min. The HPLC had an aliquot of 10 μL injected into it, and each sample was analyzed in triplicate [11]. The gradient system included solvent A (methanol: acetic acid: deionized water, 10:2:88) and solvent B (methanol: acetic acid: deionized water, 90:2:8). Different gradient programs were tested and the most effective gradient program started with 100% A at 0 min, 85% A at 5 min, 50% A at 20 min, 30% A at 25 min, and 100% B from 30 to 40 min as stated by Zeb [12]. The chromatograms were obtained using 280, 320 and 360 nm for analysis of phenolic compounds and the spectra were measured from 190 to 450 nm. The identification was carried out by the peak identification, and quantification of phenolics was carried out using the standard and established calibration curves in the concentration range of 0.333 to 1.666 ng/µL curves [12]. The linear regressions with a correlation coefficient were established between 0.9991 and 0.9996. Results for predominant phenolic acids and flavonoids were expressed in mg/kg leaves.

### 2.6. Ferric Reducing Antioxidant Power (FRAP) Assay

The FRAP assay (μmol TEAC/100 g) was performed to determine the ferric reducing antioxidant power using FRAP reagent solution (10 mmol/L TPTZ (2,4,6-tris(2-pyridyl)-1,3,5-triazine)) acidified with concentrated HCl, 20 mmol/L FeCl_3_, and nightshade leaves (0.2 g) homogenized in 2 mL CH₃COONa buffer at a pH of 3.6, as described by Managa et al. (2020) [1].

### 2.7. Proximate Analysis of the Final Product

Proximate analysis used standard methods, as reported by Managa et al. (2020) [1] without any modifications. Leaf samples (100 g) in three replicate samples were used for proximate analysis without modifications. Nitrogen content was determined using the Kjeldahl method; quantifying the nitrogen content was by converting it to protein and multiplying by a factor of 6.25. The hexane and soxhlet extraction method determined the fat content using 5 g of leaf samples. Dried ground leaf sample (2 g) was digested with 0.25 M H_2_SO_4_ and 0.3 M NaOH solution to determine the fibre content, and a dried powdered sample (5 g) was used to calculate the ash content. The estimation of the carbohydrate content used the following formula: 100 − (weight in grams (moisture + proteins + lipids + ash + fibres) in 100 g of leaves). Calorific value was calculated as (% proteins × 2.44) + (% carbohydrates × 3.57) + (% lipids × 8.37). A set of 20 mg sample mixed with 12 mL of HNO_3_ and flame photometer quantified the Na content [10].

### 2.8. Statistical Analysis

The experiments, performed in a completely randomized design, had three to five replicates per treatment and the determinations repeated twice. One-way analysis of variance (ANOVA) analyzed the significant differences between different LAB strains on different parameters at *p* < 0.05. Fisher’s protected t-test, with a least significant difference (LSD) at 5% level of significance, separated the treatment means. The statistical program Minitab for Windows (2018) analyzed the data.

## 3. Results and Discussion

### 3.1. Effect of Fermentation on Physicochemical Parameters and Ascorbic Acid Content

The initial pH value of the vegetable preparation was 6.34. The pH decreased rapidly after 24 h fermentation, irrespective of the LAB strain (Figure 1). The kinetics of the decrease differed between the two strains because the medium fermented with strain *75* reached a pH value of 5.60 after 24 h compared to 6.00 for strain *17a*. The LAB strain *56* showed the smallest decrease in pH (5.83) after 24 h of fermentation.

The results further showed that during the entire fermentation process at each incubation time point, the inoculation of strain LAB *17a* consistently lowered the pH. The starter cultures, LAB strains *75* and *17a*, reduced the pH (5.20) significantly after 72 h of fermentation. Production of various organic acids, with lactic acid as the predominant acid, could have contributed to the low pH [13]. The LAB strains produced lactic acid and CO_2_ from carbohydrates via carbohydrate metabolism pathways (Embden–Meyerhof, phosphoketolase, tagatose-6-phosphate, and Leloir pathways) and the carbonic acid produced during the metabolic pathway decreases the pH [13]. As a homofermentative bacterium, *L. plantarum* produces more lactic acid than heterofermentative *W. cibaria* [14]. Due to its lower pKa value (3.86), lactic acid acts as a stronger acidifier than the other organic acids, especially acetic acid (pKa value 4.76) [15]. The LAB fermentation helps to preserve food by making it hard for other microorganisms to thrive by decreasing the pH. Reportedly, *L. plantarum* has similar pH changes due to higher lactic acid content during fermentation of Chinese sauerkraut [16].

The TSS content was significantly reduced after 24 h of fermentation regardless of the LAB strain used (Figure 2), indicating a fast utilization of sugars. Fermentation by strains *17a* and *64* showed the highest TSS (6.87%) on day 3; the lowest TSS (6.42%) was observed after fermenting for 72 h with strain *75*. Sugars are the major soluble solids [17]. Kaprasob et al. (2017) [18] reported that the decrease in total sugar in fermented cashew apple juice was greater with *L. plantarum*, and the bacteria consumed sugar during the fermentation of this juice at a much faster rate. Reduction of TSS can be associated with bioconversion of sugar into lactic acid and consumption of sugar for the growth and metabolism of the bacteria. Similar decreases in TSS were reported during fermentation of traditional ‘snake fruit’ [19].

Fermentation with different LAB strains influenced the total color difference ∆E in nightshade leaves (Figure 3). LAB strains *17a* and *75* caused the least total color difference compared to the raw leaves. After fermentation (day 3), LAB strains *17a* and *75* resulted in pH values of 5.2, whilst the other LAB species had higher values. Acetic acid produced by the heterofermentative bacteria degraded chlorophyll *a* and *b* more than lactic acid [20]. The ∆E was significantly higher in leaves fermented with LAB strains *56* and *64* (Figure 3); this was probably due to the changes in pH during fermentation, which could have produced new compounds (Mg free chlorophyll derivatives) and carotenoid with 5,8-epoxide groups or chemical oxidation of phenolic compounds, and the production of brown pigments (*o*-quinones) may have masked the original green color of the leaves [21]. The blanching treatment of the leaves at 95 °C for 5 min probably inactivated the polyphenol oxidase enzymes.

Fermented leaves with LAB strains *75* and *17a* demonstrated the highest retention of ascorbic acid content (Figure 4). Cactus cladodes (*Opuntia ficus-indica* L.) [22] and amaranthus paste [23], showed a similar increasing trend in ascorbic acid after fermentation. In contrast, no increase in ascorbic acid was detected in pineapple juice fermented with LAB strains [24].

### 3.2. Phenolic Components and Antioxidant Activity

Table 1 illustrates the influence of fermentation by different LAB strains on the changes of different phenolic compounds in nightshade leaves. In total, there were seven phenolic acids and three flavonoids quantified. 2,5-dihydroxybenzoic, coumaric acids and catechin were the predominant phenolic acids and flavonoids, respectively, in the lactic acid fermented nightshade leaves. LAB strain *75* demonstrated a significant increase in gallic (281.0 mg/kg), vanillic (1352.0 mg/kg), coumaric (1577.0 mg/kg), ellagic (453.5 mg/kg) acids, catechin (1322.0 mg/kg) and quercetin (582.5 mg/kg) after fermentation, whilst caffeic and ferulic acids were not detected. Ferulic and caffeic acids could possibly have been decarboxylated (phenolic acid decarboxylase) to other compounds, such as 4-vinylphenol, 4-vinyl guaiacol or 4-vinyl catechol, or could have been reduced by the action of phenolic acid reductase to hydroxyphenylpropionic acids, such as dihydrocaffeic and dihydroferulic acids [25]. Similarly, *L. plantarum* GK3 decarboxylated most of the phenolic acids apart from gallic acid. Strain GK11 decarboxylated most of the phenolic acids except ferulic acid [26]. Additionally, *L. plantarum* CECT 748^T^ metabolized caeffeic, ferulic, *p*-coumaric and *m*–coumaric out of 19 food phenolic acids [27]. Decarboxylation of ferulic and caffeic acids could have affected the total polyphenol content in nightshade fermented with LAB strain *75.* The presence of vinyl derivatives at lower concentrations can provide a pleasant aroma but at higher levels, an undesirable flavor [28]. Fermentation with LAB strains *64* and *17a*, showed the highest concentrations of caffeic and ferulic acids, respectively, probably due to the inability to use or metabolise these into vinyl derivatives [25]. LAB strain *56* significantly reduced the concentration of vanillic and ferulic acids after fermentation, possibly due to bacterial decarboxylation metabolism.

The significant reduction in gallic acid content after fermenting with strain *56* was due to decarboxylation by gallic acid decarboxylase to pyragallol; accumulation of pyragallol was detected (Appendix A). The HPLC-DAD chromatogram showed lower levels of pyragallol after fermentation with strain *56* compared to *75.* Researchers have reported the activity of tannase and gallate decarboxylase in *L. plantarum* [29]. Thus, the increase in gallic and ellagic acid levels after fermentation is probably due to the tannase activity of LAB strain *75*, resulting in bioconversion of gallotanins and ellargitannins. There was no detection of ellagic acid after fermentation with strains *17a* or *64*, or in raw leaves (Table 1).

Fermentation increased the catechin content in mottled cowpea [30]. Fermentation with LAB strains increased the flavonoid concentrations by converting the complex polyphenols to simple flavonoids [31]. The concentration of total polyphenolic compounds (TPC) in the fermented nightshade product increased from 6007.8 mg/kg (raw leaves) to 8638.0 mg/kg, 8246.5 mg/kg, 8016.8 mg/kg, 5681.5 mg/kg, and 3822.5 mg/kg after fermenting with LAB strains *75*, *17a*, *64*, *21*, and *56* respectively. Similarly, fermentation with LAB strains increased the total phenolic compounds in kiwi fruit [32]. LAB strains including *L. plantarum* possess β-glucosidase enzymes that can hydrolyze the flavonoid conjugates during fermentation and influence the bioavailability of polyphenols [32]. In addition, the stability of the phenolic compounds was associated with pH, and a lower pH was reported to stabilize the pH [33]. After analyzing the above-mentioned changes related to phenolic compounds, the proposal is that overall, strain *75* enhanced accumulation of the majority of phenolic compounds other than caffeic and ferulic acids. The metabolization and transformation of the phenolic compounds differed between strain *75* and the others, probably due to the individual adaptability and ability of the strains to produce more hydrolytic enzymes [33].

This observed reduction in total phenolic compounds in nightshade leaves fermented with LAB strain *56* was probably due to detoxification and utilization of phenolic acids as a carbon source [22]. TPC showed a strong negative correlation (*r*^2^ = −0.966, *p* < 0.05) with the total color difference (∆E), which confirms the chemical oxidation of phenolic compounds and production of browning pigments (*o*-quinones) [21]. This could have been the reason for the observed reduced color difference (∆E) in leaves fermented with LAB strains *75* and *17a*.

Figure 5 presents the FRAP antioxidant activity of nightshade vegetables after fermentation. Fermentation by LAB strains *75* and *17a* increased the antioxidant activity of the nightshade leaves by 11.9% and 7.1%, respectively, compared to the unfermented (raw) leaves. Nightshade leaves fermented with strain *56* showed the lowest antioxidant activity. Coumaric acid contributed significantly to the antioxidant property followed by vanillic and gallic acids (Appendix A). Therefore, an increase in antioxidant activities was associated with the release of phenolic compounds during fermentation. However, Kaprasob et al. (2017) [18] showed a decrease in antioxidant activity in cashew apple juice and explained that this could be due to oxidation of phenolic compounds. Numerous studies have underlined a beneficial health effect for antioxidant-rich foods, such as reducing the risk of non-communicable diseases and premature ageing [34].

Each value represents the mean value of five replicate samples. Means followed by the same letter are not significantly different (*p* < 0.05).

### 3.3. Proximate Analysis

Proximate analysis was undertaken in the product showing reduced color difference and the highest antioxidant property in nightshade leaves after fermenting with LAB strains. On that basis, the leaves chosen for proximate analysis underwent fermentation with LAB strain *75*; details are in Table 2. Fat and carbohydrates were low in fermented leaves. Lower calorific carbohydrate is associated with the lower calculated energy 136.96 kJ or 32.73 cal/100 g of leaves. Since the total caloric value is less than 300 cal, fermented nightshade leaves are a low-calorie product. The final fermented product was low in protein and fiber but high in Na content. Na intake is set at 2.253 mg per day, according to the USDA dietary guidelines [35], and 0.98 g fermented nightshade leaves contributed to that portion.

## 4. Conclusions

Different LAB strains used in this study affected the concentrations of phenolic components and the antioxidant activity of nightshade leaves; however, the greater impact depends on the strain used during fermentation. LAB strains isolated from fermented vegetables exhibit numerous functional properties [36,37,38,39,40]. Among those, *L. plantarum* is the species that has been focused on in most studies. The ability of this bacterium to modulate antioxidant activity and phenolic compound composition appears strain-dependent, but also dependent on the matrix investigated. *L. plantarum* strain *75* is a potential starter culture for the improvement of phenolic composition in nightshade leaves compared to the other strains, and it enhanced the use of nightshade leaves as a functional ingredient or food. Since increased concentrations of phenolic compounds can affect the astringency or bitterness of nightshade leaves, sensory evaluation is necessary in the future. Phenolic compounds are associated with health benefits, such as anti-diabetic activity, and performing biological activities after fermenting the nightshade leaves with *L. plantarum 75* shows their beneficial effects.

## Figures and Tables

**Figure 1 microorganisms-08-01324-f001:**
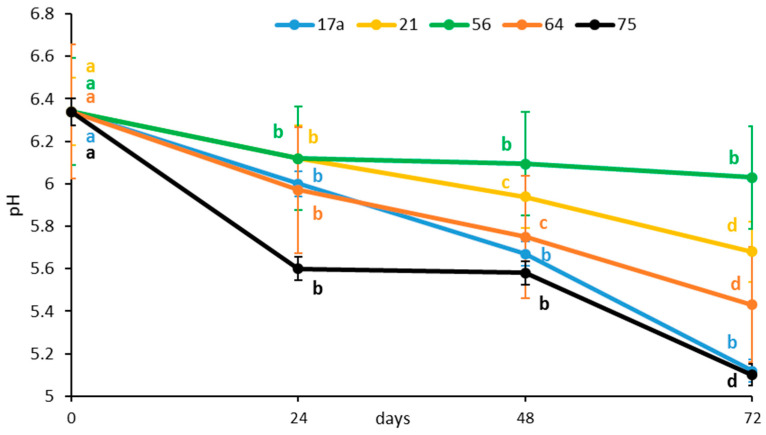
Influence of lactic acid fermentation with different LAB strains on pH of the brine solution of African nightshade (*Solanum retroflexum*) leaves. Each value represents the mean value of three replicate samples. Means followed by the same letter within a series are not significantly different (*p* < 0.05).

**Figure 2 microorganisms-08-01324-f002:**
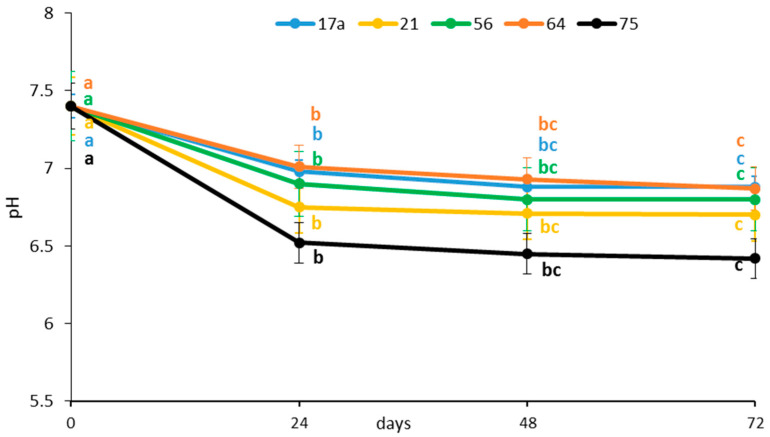
Influence of lactic acid fermentation with different LAB strains on total soluble solids of the brine solution of African nightshade (*Solanum retroflexum*) leaves. Each value represents the mean value of three replicate samples. Means followed by the same letter within a series are not significantly different (*p* < 0.05).

**Figure 3 microorganisms-08-01324-f003:**
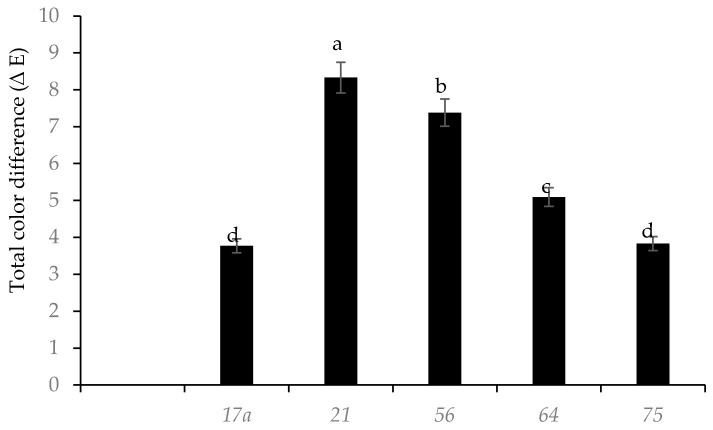
Influence of lactic acid fermentation with different LAB strains on color difference compared to raw (*Solanum retroflexum*) leaves. Each value represents the mean value of five replicate samples. Means followed by the same letter are not significantly different (*p* < 0.05).

**Figure 4 microorganisms-08-01324-f004:**
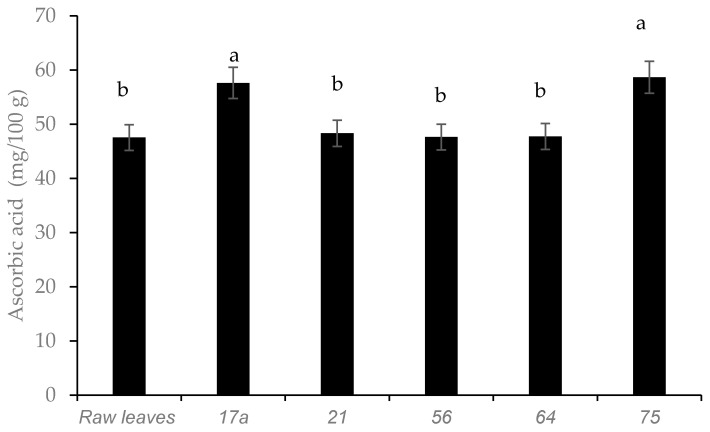
Influence of lactic acid fermentation with different LAB strains on ascorbic acid content in African nightshade (*Solanum retroflexum*) leaves. Each value represents the mean value of five replicate samples. Means followed by the same letter are not significantly different (*p* < 0.05).

**Figure 5 microorganisms-08-01324-f005:**
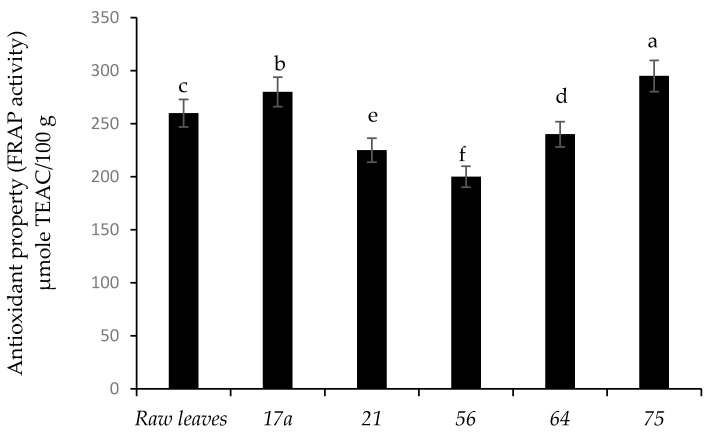
Influence of lactic acid fermentation of *Lactobacillus* strains on antioxidant activity (FRAP activity) of African nightshade leaves (*Solanum retroflexum*).

**Table 1 microorganisms-08-01324-t001:** Influence of fermentation by different Lactobacillus strains on phenolic components of African nightshade (*Solanum retroflexum*) leaves.

				LAB Strain				
**Phenolic acids (mg/kg)**		RT (min)	Raw leaves	*17a*	*56*	*64*	*21*	*75*
Gallic acid	C_7_H_6_OH_5_	6.78	162.8 ± 1.1 ^d^	178.0 ± 1.3 ^c^	150.5 ± 1.09 ^e^	171.5 ± 1.4 ^cd^	210.0 ± 1.7 ^b^	281.0 ± 1.2 ^a^
Caffeic acid	C_9_H_8_O_4_	20.55	360.5 ± 1.4 ^b^	294 ± 0.98 ^c^	77.5 ± 0.21 ^d^	2145 ± 1.4 ^a^	395 ± 1.9 ^b^	nd
Vanillic acid	C_8_H_8_O_4_	18.70	403.0 ± 0.5 ^b^	343 ± 2.3 ^c^	21.0 ± 1.7 ^e^	383.5 ± 0.67 ^b^	245 ± 0.98 ^d^	1352.0 ± 1.23 ^a^
2,5 Dihydroxybenzoic acid	C_7_H_6_O_4_	14.50	1593 ± 1.5 ^c^	2406 ± 0.7 ^b^	489 ± 1.8^f^	1003.3 ± 2.2 ^e^	1267 ± 1.2 ^d^	2827.5 ± 1.4 ^a^
p-Coumaric acid	C_9_H_8_O_3_	49.37	1115 ± 0.76 ^d^	1269.5 ± 1.8 ^c^	929 ± 1.8 ^e^	1387.0 ± 1.5 ^b^	980.5 ± 2.1 ^e^	1577 ± 0.56 ^a^
Ferulic acid	C_10_H_10_O_4_	20.61	1321 ± 9.8 ^d^	2343.0 ± 1.2 ^a^	1030 ± 1.6 ^e^	2000 ± 1.9 ^b^	1487 ± 1.6 ^c^	nd
Ellagic acid	C_14_H_6_O_8_	12.3	nd	nd	113.0 ± 1.4 ^c^	nd	191.0 ± 1.6 ^b^	453.5 ± 1.8 ^a^
**Total phenolic acids**			4955.3 ± 1.23 ^d^	6833.5 ± 1.41 ^b^	2810 ± 1.12 ^e^	7090.3 ± 1.70 ^a^	4775.5 ± 1.63 ^d^	6491 ± 1.16 ^c^
Flavonoids (mg/kg)								
Catechin	C_15_H_14_O_6_	12.75	726.5 ± 2.5 ^b^	658.5 ± 1.3 ^cd^	574.5 ± 0.78 ^d^	669 ± 1.3 ^bc^	602.0 ± 0.45 ^c^	1322.0 ± 1.7 ^a^
Quercetin	C_15_H_10_O_7_	62.34	326.0 ± 1.4 ^d^	498.5 ± 1.8 ^b^	438.0 ± 1.7 ^c^	116.0 ± 1.4^f^	304.0 ± 1.1 ^e^	582.5 ± 1.7 ^a^
Luteolin	C_15_H_10_O_6_	20.00	nd	256 ± 1.5 ^a^	nd	141.5 ± 2.1 ^c^	nd	242.5 ± 2.4 ^b^
**Total flavonoids**			1052.5 ± 1.21 ^c^	1413.0 ± 1.83 ^b^	1012.5 ± 1.62 ^c^	926.5 ± 0.95 ^d^	906.0 ± 1.86 ^d^	2147.0 ± 1.40 ^a^
**Total polyphenol content**			6007.8 ± 1.34 ^c^	8246.5 ± 1.50 ^b^	3822.5 ± 1.75 ^d^	8016.8 ^b^	5681.5 ^c^	8638 ± 1.69 ^a^

Each value represents the mean value of five replicate samples. Means followed by the same letter within the row for a specific parameter are not significantly different (*p* < 0.05); nd-not detected.

**Table 2 microorganisms-08-01324-t002:** Proximate analysis data of the nightshade leaves fermented with *L. plantarum* strain *75*.

Component	Per Fermented Vegetable (100 g Fresh Weight)
Energy (kJ)	136.96
Moisture (g)	81.51 ± 0.22 *
Fat (g)	0.23 ± 0.03
Protein (g)	3.86 ± 0.19
Carbohydrate (g)	2.51 ± 0.05
Fibre (g)	2.52 ± 0.30
Ash (g)	9.91 ± 0.50
Na (sodium) (mg)	231.00 ± 2.81

* Standard deviation, *n* = 3.

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
