# Peer review of "Effect of Lactic Acid Fermentation on Color, Phenolic Compounds and Antioxidant Activity in African Nightshade"

_microorganisms, 2020, doi:10.3390/microorganisms8091324_

Round 1

Reviewer 1 Report

The presented investigation refers to a treatment of African nightshade by LAB, inducing lactic acid fermentation. As the leaves of this plant are widely used as a food, the authors’ state that subjected to fermentation, leaves have improved healthy qualities and became functional food.

Although the article is concentrated on some beneficial features of LA fermentation in this case, like changes in phenolic content and antioxidant activity of the fermented product, some important data should be added. Lactic acid is the main consequence of LAB treatment and the main cause of benefits of the process, so LA quantification is necessary to add. The measurement of pH decrease is insufficient. Likewise, there is no information about the fermented sugars or polysaccharides of the leaves.

Some minor points:

Table 1 is unnecessary.

The quality of the Figures should be improved. For instance, a-d symbols are not described in the legend and overlap each other.

Line 132 – syntax error – “605 fibers”.

To cite the source pointing out that lactic acid decreases pH is unnecessary. On the other hand, the cited literature is very poor, especially about LA fermentation of indigenous foods and beverages by L. plantarum. For example: Appl. Environ. Microbiol. 1994, 60, 4319–4323; Nutrients, 2020, 12, 1118; Food Res. Int. 2016, 89, 1095–1105; Int. J. Food Microbiol. 2003, 80, 201–210, etc.

Author Response

REVIEWER 1

The presented investigation refers to a treatment of African nightshade by LAB, inducing lactic acid fermentation. As the leaves of this plant are widely used as a food, the authors’ state that subjected to fermentation, leaves have improved healthy qualities and became functional food.

Although the article is concentrated on some beneficial features of LA fermentation in this case, like changes in phenolic content and antioxidant activity of the fermented product, some important data should be added. Lactic acid is the main consequence of LAB treatment and the main cause of benefits of the process, so LA quantification is necessary to add. The measurement of pH decrease is insufficient. Likewise, there is no information about the fermented sugars or polysaccharides of the leaves.

We thank the reviewer for his suggestions of improvement of our manuscript.

Lactic acid bacterium metabolism of sugars is well-known. We agree that the production of organic acids is the main consequence of lactic acid fermentation and contributes to increased shelf-life of the produced foods. However, this in not the main beneficial effect from a nutritional point of view. As pointed by several studies (see below) phytochemicals, and especially phenolic compounds, are the main nutritional beneficial characteristics of nightshade.

If the decrease of sugar content is an important nutritional feature for fermented fruit, which contain high sugar levels, this is not the case for vegetables. We report a decrease in TSS which indicates an efficient consumption of sugars during fermentation (L. 175 and following). In fermented nightshade (Solanum retroflexum), we report a carbohydrate level of 2.51 g/100g fresh weight, which is low compared to most fruit.

Kirigia, D., Winkelmann, T., Kasili, R., Mibus, H., 2019. Nutritional composition in African nightshade (Solanum scabrum) influenced by harvesting methods, age and storage conditions. Postharvest Biology and Technology 153, 142–151. https://doi.org/10.1016/j.postharvbio.2019.03.019

Managa, M.G., Mpai, S., Remize, F., Garcia, C., Sivakumar, D., 2020. Impact of moist cooking methods on colour, anti-nutritive compounds and phenolic metabolites in African nightshade (Solanum retroflexum Dun.). Food Chemistry 325, 126805. https://doi.org/10.1016/j.foodchem.2020.126805

Odongo, G.A., Schlotz, N., Baldermann, S., Neugart, S., Huyskens-Keil, S., Ngwene, B., Trierweiler, B., Schreiner, M., Lamy, E., 2018. African Nightshade (Solanum scabrum Mill.): Impact of Cultivation and Plant Processing on Its Health Promoting Potential as Determined in a Human Liver Cell Model. Nutrients 10. https://doi.org/10.3390/nu10101532

We modified the text l. 170-166 accordingly.

Some minor points:

Table 1 is unnecessary.

Table 1 was removed, and the text modified to specify the origin of strains. Actually, the plant (not dairy) origin of strains is an important feature regarding their use to ferment vegetables.

The quality of the Figures should be improved. For instance, a-d symbols are not described in the legend and overlap each other.

The figures were modified accordingly.

Line 132 – syntax error – “605 fibers”.

Thank you. This was corrected.

To cite the source pointing out that lactic acid decreases pH is unnecessary. On the other hand, the cited literature is very poor, especially about LA fermentation of indigenous foods and beverages by L. plantarum. For example: Appl. Environ. Microbiol. 1994, 60, 4319–4323; Nutrients, 2020, 12, 1118; Food Res. Int. 2016, 89, 1095–1105; Int. J. Food Microbiol. 2003, 80, 201–210, etc.

We modified the conclusion and added some references (37 to 41) regarding lactic acid fermentation of vegetables. We did not add the proposed references as they are related to cereal-based foods.

Reviewer 2 Report

This is an interesting scientific research for industrials and scientists.The research studied the influence of fermentation at 37°C during 3 days, by different  lactic acid bacteria strains (Lactobacillus plantarum , Weissella cibaria , Leuconostoc  pseudomesenteroides , W. cibaria and Lactobacillus plantarum upon organoleptic proprieties such as color, pH, total soluble solids (TSS), phenolic compounds and antioxidant activity of African nightshade (leaves). 

Appropriate methodology was used for pH measurement (pH meter),color difference(colorimeter) and TSS(refractometer).The quantification of phenolic compounds and flavonoids was done by using HPLC-DAD .FRAP assay was applied to determine the ferric reducing antioxidant power and finally proximate analysis of the leaf samples was performed with classic methodology.The fermentation impact is strictly depending on the strain used during fermentation. L. plantarum strain 75 is a potential starter culture for the improvement of  phenolic composition in nightshade leaves compared to the other strains, and it enhanced the use of  nightshade leaves as a functional ingredient or food due to the health benefits of phenolics. However ,as phenolic  compounds can affect the astringency or bitterness of nightshade leaves, a sensory evaluation of the final product  should be necessary.

It is a well written paper with appropriate methodology ans should be of intest to scientists.

my suggestion is to ACCEPT and  publish the paper in its present form.

Author Response

We thank the reviewer for careful reading and assessment of our work.

Round 2

Reviewer 1 Report

Only one little thing - MRS is an abbreviation of "de Man, Rogosa...", instead "Man,...".